# OpenReview forum: "Image Hijacks: Adversarial Images can Control Generative Models at Runtime"
_ICLR.cc/2024/Conference — Submitted to ICLR 2024_

### Official Review · Reviewer_dgQe · 2023-10-20

**Soundness:** 2 fair
**Presentation:** 3 good
**Contribution:** 2 fair
**Rating:** 5
**Confidence:** 4

**Summary:**

This paper studies adversarial images (referred to as image hijacks) in the context of attacking large vision-language models (VLMs). Specifically, the authors explore three types of attacks, i.e., specific string attack, leak context attack, and jailbreak attack, which target three different undesirable behaviours of VLMs. Three different image constraints are considered: Lp norm, stationary patch, and moving patch.

**Strengths:**

- "Adversarial images on VLMs" is an interesting and promising topic.
- The paper is well written, including sufficient example visualizations and clearly described technical details.
- Real-world experiments with human studies are conducted, involving both the tests with drawing and pasting tapes.

**Weaknesses:**

- The experimental studies in this paper are limited. Only the ideal, white-box setting is considered. The authors should follow recent attempts (e.g., the GCG paper) to explore more realistic attack settings, such as query-based and transfer-based attacks. Moreover, although the authors envision three different undesired behaviors, the implemented three attacks are indeed following the same attack goal which is to output a pre-defined text.  More diverse attack goals should be explored.

- Considering this paper studies a really hot topic, it should highlight its specific contributions, especially in terms of technical novelty, compared to other (concurrent) similar studies. In the current version, only the differences in attack settings are briefly mentioned in Section 5.

- The reviewer does not get the idea of splitting the dataset into train, val, and test sets. Is the perturbation vector universal, i.e., it is trained over a lot of original images and once trained can be directly applied to any testing images? If this is the case, it should be explicitly mentioned in the paper. Otherwise, it is confusing that creating adversarial images requires image (or model) training. In addition, why didn’t the authors explore the common setting of image-specific attacks?

- Since the baseline GCG attack was also designed to achieve “exact match”, it is reasonable to compare with it also on the rest two types of attacks: specific string and leak context.

- Figure 4 takes an unreasonably large space (i.e., one page) given the fact those three considered types of constraints are well-known in the literature.

**Questions:**

See the above weaknesses.

---

> ### Author Response · Authors · 2023-11-21
> **Author Response to Reviewer dgQe**
>
> We thank the reviewer for their detailed review. Below, we respond to each of the concerns they raised. **Could the reviewer please let us know if this addressed their concerns?**
>
> >“it should highlight its specific contributions, especially in terms of technical novelty, compared to other (concurrent) similar studies”
>
> We have added Table 3, which highlights our specific contributions, in Appendix C.
>
> In our [OpenReview comment to all reviewers](https://openreview.net/forum?id=ucMRo9IIC1&noteId=8Eu4wFSlNR), we give a description of further changes to increase the novelty of our work. In summary, we draw your attention to following changes:
>
>
> 1. We have moved the prompt matching algorithm from the Appendix to Section 5. This is an **entirely nove**l adversarial attack method that is different to anything presented by Qi et al. and Carlini et al. The prompt matching algorithm we introduce allows for the creation of adversarial images that induce similar model behaviors to textual prompts. This means we can create attacks that are **hard to encapsulate in training datasets**.
> 2. Since our initial submission, we have added a more comprehensive comparison to text-based adversarial attacks. Because Qi et al. and Carlini et al. are restricted to jailbreak attacks, they only show that an existing text vulnerability (jailbreaks) can also be done with images. We have added results with a state-of-the-art text-based attack on the specific string and leak-context attacks. We find that in this broader setting, state-of-the-art text attacks fail; for example, text attacks get 0% success rate on the leak context attack. Thus, our study is the first to show **new image attacks that were impossible with just text**.
> 3. As you noted, all related works released after May 28, 2023 are concurrent as per [ICLR guidelines](https://iclr.cc/Conferences/2024/ReviewerGuide).
>
> >“The implemented three attacks are indeed following the same attack goal which is to output a pre-defined text.”
>
> It is false that our three attacks are all outputting a predefined text. For each of the three different attacks, the output is defined in a different way.
> * The specific string attack is a predefined text. The output of the model only depends on this text.
> * In the leak context attack, the model's output depends on the user's input. The attack is only successful if the user's input appears in the output.
> * In the jailbreak attack, the attack depends on the model's RLHF safety training. The definition of a successful attack is if the output violates the model's safety training. It does not depend on any predefined text.
>
>
> >“Only the ideal, white-box setting is considered”
>
> We agree this is a limitation of our work, and we have added a discussion about it to Section 7. Nevertheless, we see our work being broadly relevant in two ways. First, it shows the existence of vulnerabilities in VLMs, even if the field has yet to develop black-box attacks that can find these vulnerabilities. Second, many apps are currently being developed with open-source models, and we expect this trend only to increase in the future. Our work is directly relevant to any app built using an open-source VLM.
>
> >“The reviewer does not get the idea of splitting the dataset into train, val, and test sets”
>
> A single adversarial image is trained on a dataset of textual prompts to ensure that the image induces the desired model behavior **no matter the user’s input text**. We take a dataset of possible user inputs and split it into training, val, and test sets. We have modified Section 2 to help make this more clear.
>
> >“Since the baseline GCG attack was also designed to achieve “exact match”, it is reasonable to compare with it also on the rest two types of attacks: specific string and leak context.”
>
> **We have added GCG baseline numbers for all attack types**. We find GCG underperforms image adversaries significantly. In particular, the GCG attack is unable to get any success with the leak context attack, compared to an optimal success rate of 96% for image based attacks. This finding is a novel contribution of our paper: **image-based attacks open up the possibility for new attacks that are impossible with existing text-based attacks**.
>
> > “figure 4 takes an unreasonably large space”
>
> We have reduced figure 4’s size and moved it to the Appendix.

---

> ### Author Response · Authors · 2023-11-23
> **Author Response to Reviewer dgQe - New Black-Box Transfer Experiments**
>
> **We have added black-box transferability experiments to Appendix E**. We try two settings: directly inputting our original attacks into new models, and also training on an ensemble of models to try improving transferability. We get 0% success rate in both cases, establishing preliminary negative results for black-box transferability. (As with any negative empirical result, it will be important for future work to continue investigating this question.)
>
> Please see our [top-level comment](https://openreview.net/forum?id=ucMRo9IIC1&noteId=e7y7rRoDly) and Appendix E for more details.

---

### Official Review · Reviewer_DvbG · 2023-10-30

**Soundness:** 3 good
**Presentation:** 3 good
**Contribution:** 2 fair
**Rating:** 5
**Confidence:** 4

**Summary:**

This paper studies the security problems of Vision-Language Models (VLMs), and propose to add unnoticeable perturbations onto the image inputs, to mislead the model have various types of adversarial behaviors.

**Strengths:**

The proposed attacks absolutely make sense, and demonstrate the weakness, the security risks of the VLMs.

**Weaknesses:**

The greatest concern is that the paper does not study the transferability behavior of the proposed attacks to close-source VLMs. Although the authors mention that the jailbreak attack in [Zou 2023] did similar experiments to transfer the attacks from open-source LLMs to close-source LLMs, it can not ensure the VLM attacks can also have the ability to transfer. If this part of experiment is not provided, it would dramatically limit the significant of the contribution of this paper.

Beyond the study of transferability, I found the methodology is not significantly novel, compared to existing attack methods such as Jailbreak attacks in LLMs and image adversarial examples. Therefore, without the transferability study, this work seems solely a white-box attack paper without significant novelty, because people already know that the DNN models are vulnerable to adversarial attacks.

**Questions:**

Plz see the weakness part.

---

> ### Author Response · Authors · 2023-11-21
> **Author Response to Reviewer DvbG**
>
> We thank the reviewer for their review, and respond to each of the concerns they raised below:
>
> >“the methodology is not significantly novel”
>
> Please see our [comment above to all reviewers](https://openreview.net/forum?id=ucMRo9IIC1&noteId=8Eu4wFSlNR), which discusses the technical contribution and novelty of the paper. In summary, we draw your attention to following changes:
>
> 1. All the related work mentioned by reviewers was released after May 28, 2023. Per [ICLR guidelines](https://iclr.cc/Conferences/2024/ReviewerGuide), these papers are concurrent with our own.
> 2. We have moved the prompt matching algorithm from the Appendix to Section 5. This is an **entirely nove**l adversarial attack method that is different to anything presented by Qi et al. and Carlini et al. The prompt matching algorithm we introduce allows for the creation of adversarial images that induce similar model behaviors to textual prompts. This means we can create attacks that are **hard to encapsulate in training datasets**.
> 3. Since our initial submission, we have added a more comprehensive comparison to text-based adversarial attacks. Because Qi et al. and Carlini et al. are restricted to jailbreak attacks, they only show that an existing text vulnerability (jailbreaks) can also be done with images. We have added results with a state-of-the-art text-based attack on the specific string and leak-context attacks. We find that in this broader setting, state-of-the-art text attacks fail; for example, text attacks get 0% success rate on the leak context attack. Thus, our study is the first to show **new image attacks that were impossible with just text**.
>
> >“paper does not study the transferability behavior of the proposed attacks to close-source VLMs”
>
> We agree this is a limitation of our work, and we have added a discussion about it to Section 7. Nevertheless, we see our work being broadly relevant in two ways. First, it shows the existence of vulnerabilities in VLMs, even if the field has yet to develop black-box attacks that can find these vulnerabilities. Second, many apps are currently being developed with open-source models, and we expect this trend only to increase in the future. Our work is directly relevant to any app built using an open-source VLM.
>
> Note also that GPT-4V (mentioned by NiWH) was released after ICLR’s abstract submission deadline.
>
> **Have we addressed the reviewer's concerns?**

---

> ### Comment · Reviewer_DvbG · 2023-11-21
>
> Thanks for the response. Due to the lack of transferability experiment, I would keep my original rating.
>
> Other concerns still remains. For example, the revised version has a weird structure. If the major novelty is about "prompt matching", this should be introduced in more details in early sections, instead of in the experiment section. Besides, I hope the authors could try avoiding provide irrelevant information which are not asked in this review.

---

> ### Author Response · Authors · 2023-11-23
> **Author Response to Reviewer DvbG Official Comment - New Black-Box transfer experiments**
>
> **We have added black-box transferability experiments to Appendix E**. We try two settings: directly inputting our original attacks into new models, and also training on an ensemble of models to try improving transferability. We get 0% success rate in both cases, establishing preliminary negative results for black-box transferability. (As with any negative empirical result, it will be important for future work to continue investigating this question.)
>
> Please see our [top-level comment](https://openreview.net/forum?id=ucMRo9IIC1&noteId=e7y7rRoDly) and Appendix E for more details.

---

> ### Comment · Reviewer_DvbG · 2023-11-23
>
> Thanks for the additional result provided. I really appreciate authors' effort to provide the result on the transferabilty study.
>
> From my perspective, it is critical for the attack to succeed in black box models, which is also agreed by other reviewers. Based on the result provided by the author, it demonstrates the possibility, although the attack is not achieved eventually. Therefore, a lot of more interesting studies can be done to improve the paper, such as trying to raise the perturbation budget or modify the attack objective.
>
> As a conclusion, I hesitate that this study is completed on the current stage, and I hope the authors can provide a more comprehensive study on the transferability in the future revision.

---

### Official Review · Reviewer_NiWH · 2023-10-31

**Soundness:** 3 good
**Presentation:** 3 good
**Contribution:** 2 fair
**Rating:** 5
**Confidence:** 5

**Summary:**

The paper delves into the security aspects of vision-language models (VLMs) and their susceptibility to adversarial attacks via the image channel. It introduces "image hijacks," adversarial images that can manipulate generative models in real time. Using a method called Behavior Matching, the authors showcase three attack types: specific string attacks, leak context attacks, and jailbreak attacks. In their evaluation, these attacks have a success rate exceeding 90% on leading VLMs, with the attacks being automated and necessitating only minor image alterations. They underscore the security vulnerabilities of foundational models, hinting that countering image hijacks might be as formidable as defending against adversarial examples in image categorization.

**Strengths:**

1. Problem Motivation: The paper studies a highly pertinent and timely issue, highlighting the security vulnerabilities of VLMs, which are increasingly being used in various applications.

2. Methodology: The behavior-matching method for creating image hijacks is sound and intuitive. The three types of attacks presented provide a wide understanding of the potential threats.

3. Evaluating the Evaluation: The evaluation is highlighted with alarming results, with all attack types achieving a success rate above 90% against LLaVA. This supports the paper's claims about the vulnerabilities of open-sourced VLMs.

**Weaknesses:**

1. Technical Contribution: Limited novelty due to similarities with prior works (e.g., Qi et al., Carlini et al.).

2. Problem Formulation: Restricted impact concerning closed-sourced VLMs or those not directly interfacing the image channel with the hidden space.

3. Transferability of Attacks: Insufficient discussion on attack applicability to unknown VLMs accessible only via APIs (balckbox attacks, which can be a great contribution to set the difference to existing efforts).

4. Lack of Defense Discussion: No mention of potential defenses or mitigation strategies against the proposed attacks.

**Questions:**

- Technical Contribution and Novelty: While your paper introduces the specific string attack and leak context attack, which were not covered by Qi et al. and Carlini et al., the attack process appears to be highly similar to these prior works. Could you elaborate on the distinct technical innovations that differentiate your methods from these existing studies? Additionally, are there areas where you foresee further improvements or refinements to enhance the technical novelty of your approach?

- Problem Formulation: Considering the limited applicability of your problem formulation to closed-sourced VLMs or those VLMs that do not directly interface the image channel with the hidden space, how do you envision the broader relevance of your proposed attacks? Are there specific scenarios or VLM architectures where your attacks would be particularly effective?

- Transferability of Attacks: Could you expand on the transferability of your attacks to unknown VLMs, especially those only accessible via APIs, such as GPT-4V? Given that GPT-4V's technical report has already highlighted potential risks associated with the image channel and proposed baseline defenses, how do you anticipate your attacks would perform in such contexts?

- Lack of Defense Discussion: Why was there an omission of defenses or potential mitigation strategies against the proposed attacks in your paper? A discussion on potential countermeasures would enhance the paper's depth and practical relevance. Do you have insights or preliminary findings on how one might defend against the attacks you've introduced?

---

> ### Author Response · Authors · 2023-11-21
> **Author Response to Reviewer NiWH**
>
> We thank the reviewer for their detailed review, and respond to each of their questions below.
>
> >“Technical Contribution and Novelty”
>
> Please see our [comment above to all reviewers](https://openreview.net/forum?id=ucMRo9IIC1&noteId=8Eu4wFSlNR), which discusses the technical contribution and novelty of the paper. In summary, we draw your attention to following changes and comparisons to Qi et al. and Carlini et al.:
>
> 1. The works by Qi et al. and Carlini et al. were released after May 28, 2023. Per [ICLR guidelines](https://iclr.cc/Conferences/2024/ReviewerGuide), these papers are concurrent with our own.
> 2. We have moved the prompt matching algorithm from the Appendix to Section 5. This is an **entirely novel** adversarial attack method that is different to anything presented by Qi et al. and Carlini et al. The prompt matching algorithm we introduce allows for the creation of adversarial images that induce similar model behaviors to textual prompts. This means we can create attacks that are **hard to encapsulate in training datasets**.
> 2. Since our initial submission, we have added a more comprehensive comparison to text-based adversarial attacks. Because Qi et al. and Carlini et al. are restricted to jailbreak attacks, they only show that an existing text vulnerability (jailbreaks) can also be done with images. We have added results with a state-of-the-art text-based attack on the specific string and leak-context attacks. We find that in this broader setting, state-of-the-art text attacks fail; for example, text attacks get 0% success rate on the leak context attack. Thus, our study is the first to show **new image attacks that were impossible with just text**.
>
> >“Limited applicability of your problem formulation to closed-sourced VLMs”
>
> We see our work being broadly relevant in two ways. First, it shows the existence of vulnerabilities in VLMs, even if the field has yet to develop black-box attacks that can find these vulnerabilities. Second, many apps are currently being developed with open-source models, and we expect this trend only to increase in the future. Our work is directly relevant to any app built using an open-source VLM.
>
> On internal testing, we found our attacks were in no way limited to the LLaVA series of VLMs. We were also able to apply attacks to the BLIP-2 and InstructBLIP models. This is especially notable because the BLIP-2 model uses an encoder-decoder language model, different to the decoder-only language model used by LLaVA.
>
> >“Transferability of Attacks”
>
> We have added a paragraph **discussing this limitation in Section 7** of the paper. We agree with the reviewer that transferring our attacks to closed-source models would improve the impact of the paper. Upon further investigation we find that our adversarial images do not transfer to closed source models such as GPT4-V (that could be in part due to OpenAI uses defenses, such as image preprocessing, that we are unaware of and did not train our attacks to be robust to). We note that GPT4-V was only released publicly on September 25th, two days after the ICLR abstract submission deadline. We are however excited about future work that could build on our method to generate transferable attacks.
>
> >“Lack of Defense Discussion”
>
> Adversarial defenses are a particularly fraught topic, where defenses are commonly created but quickly broken with minor tweaks to the attacking algorithm (see, for example, [“Obfuscated Gradients Give a False Sense of Security.”](https://arxiv.org/abs/1802.00420)) Because one must be very careful when proposing defense algorithms, we leave this topic to future work. Still, we agree that a preliminary discussion of defenses is a valuable addition to our paper. **We have added a new paragraph discussing this topic in Section 7**.
>
> **Have we addressed the reviewer's concerns?**

---

> ### Author Response · Authors · 2023-11-23
> **Author Response to Reviewer NiWH - New Black-Box Transfer Experiments**
>
> **We have added black-box transferability experiments to Appendix E**. We try two settings: directly inputting our original attacks into new models, and also training on an ensemble of models to try improving transferability. We get 0% success rate in both cases, establishing preliminary negative results for black-box transferability. (As with any negative empirical result, it will be important for future work to continue investigating this question.)
>
> Please see our [top-level comment](https://openreview.net/forum?id=ucMRo9IIC1&noteId=e7y7rRoDly) and Appendix E for more details.

---

### Official Review · Reviewer_C2M9 · 2023-11-01

**Soundness:** 3 good
**Presentation:** 3 good
**Contribution:** 3 good
**Rating:** 5
**Confidence:** 3

**Summary:**

This paper is well-organized and easy to follow. This idea is simple yet effective.
The authors introduce the concept of image hijacks – adversarial images that manipulate the behavior of VLMs during inference time. They propose a behavior matching algorithm to train these models in a more robust manner against user input. The authors have developed three types of image hijacks: specific string attacks, jailbreak attacks, and leak-context attacks. The optimized perturbation is both imperceptible and effective.
The experiments are comprehensive, and the overall ASR is high, demonstrating the effectiveness of these methods.

**Strengths:**

+ The paper presents a rather interesting approach. Utilizing an image to replace the prompt templates of jailbreak attacks seems reasonable. As large multi-modal models continue to develop, such attack methods introduce a new area of adversarial attacks that can be better optimized using computer vision gradient information. I found the idea and the overall paper to be quite enjoyable.

**Weaknesses:**

- This paper is well-organized and easy to follow, but there is one important baseline missing: "Visual Adversarial Examples Jailbreak Aligned Large Language Models." This paper demonstrates that by using a preset adversarial sample image, a safely aligned LLM can be successfully jailbroken during subsequent model risk assessments, leading the LLM to generate harmful content. The primary attack method involves optimizing the adversarial sample image to create a specific mapping relationship between the image and malicious text.

- I suggest further research on using a single image to match a series of jailbreak attack prompt templates, such as the "Do Anything Now" (DAN) series.

- In my opinion, the patch-level attack or l-p norm attack may not be particularly meaningful. They are simply different methods for generating adversarial examples. What truly matters is the effectiveness of the mapping between adversarial examples and various hijack risks, such as specific string attacks, jailbreak attacks, and leak-context attacks.

**Questions:**

Refer to weakness.

**Details Of Ethics Concerns:**

This paper investigates the use of images to hijack large language models, which could potentially lead to the generation of toxic content in the resulting output. However, the paper overall does not present any ethical issues, and the proposed method appears to work effectively.

---

> ### Author Response · Authors · 2023-11-21
> **Author Response to Reviewer C2M9**
>
> We thank the reviewer for their detailed and insightful comments. Below, we summarize what we identify as the crucial issues raised in the review, and describe how we have addressed each. **Does this fully address the reviewer’s concerns?**
>
> > “I suggest further research on using a single image to match a series of jailbreak attack prompt templates, such as the "Do Anything Now" (DAN) series.”
>
> We added a new section of the paper, Section 5, that creates images matching arbitrary text prompts through the novel prompt matching algorithm. Naively, creating images that elicit the same model behavior as text prompts can be done by training the embeddings of images to match the embeddings of a text prompt. We find this is impossible to do because of the modality gap in multi-modal embedding spaces [1]. Instead, we develop the novel prompt matching algorithm. For details of this algorithm, see the new section 5 of the paper.
>
> Because we are already able to achieve high jailbreak rates with our prior attack methods, we use our prompt matching algorithm to conduct a new disinformation attack, editing the model’s factual recall. In our experiments, we get the model to believe the Eiffel Tower has been moved to Rome, _without training on any questions specifically chosen to be about the Eiffel Tower or Rome_.
>
> > “one important baseline missing”
>
> The method outlined by Qi et al. is concurrent work and very similar to our own method. The algorithm used is essentially the same as our own, besides using a different training set for their jailbreak attack. We accordingly omit the baseline comparison as both methods would achieve the same results if using the same training dataset and hyperparameters. We note however that we explore a far wider range of attacks than Qi et al. who only explore jailbreaking. We show a critical fact they do not: that image adversaries can **fully control the output of VLMs**.
>
> We also note the [official ICLR policy](https://iclr.cc/Conferences/2024/ReviewerGuide): “**if a paper was published (i.e., at a peer-reviewed venue) on or after May 28, 2023, authors are not required to compare their own work to that paper**”. Qi et al.’s work was first arXived Jun 22 (and has never appeared in a peer-reviewed venue).
>
> > “patch-level attack or l-p norm attack may not be particularly meaningful”
>
> We believe it is important to include the study under these constraints, to illustrate how these attacks could be deployed under credible threat models. For example, an attacker may only be able to place an image on a website and want to attack a VLM that analyzes screenshots of a user's display. Under this threat model, Image Hijacks are only of concern if they can be trained as patches on otherwise unperturbed images.
>
> [1] Mind the Gap: Understanding the Modality Gap in Multi-modal Contrastive Representation. Learning Weixin Liang*, Yuhui Zhang*, Yongchan Kwon*, Serena Yeung, James Zou. NeurIPS (2022).

---

> ### Author Response · Authors · 2023-11-23
> **Author Response to Reviewer C2M9 - New Black-Box Transfer Experiments**
>
> **We have added black-box transferability experiments to Appendix E**. We try two settings: directly inputting our original attacks into new models, and also training on an ensemble of models to try improving transferability. We get 0% success rate in both cases, establishing preliminary negative results for black-box transferability. (As with any negative empirical result, it will be important for future work to continue investigating this question.)
>
> Please see our [top-level comment](https://openreview.net/forum?id=ucMRo9IIC1&noteId=e7y7rRoDly) and Appendix E for more details.

---

### Author Response · Authors · 2023-11-21
**Author Response (2/2)**

We also note the [official ICLR policy](https://iclr.cc/Conferences/2024/ReviewerGuide): “**if a paper was published (i.e., at a peer-reviewed venue) on or after May 28, 2023, authors are not required to compare their own work to that paper**”. All of the above works are only preprints, were published after May 28, 2023, or both. All of these works are therefore concurrent with our own (including Qi et al. and Carlini et al. that were mentioned by NiWH and C2M9).


### **Attacking White-Box Models**

We agree with the reviewers that transferring our attacks to closed sourced VLMs such as GPT-4V (mentioned by NiWH) would improve the contributions of the paper. (Note though that GPT-4V was only released publicly on September 25th, less than one week before the paper submission deadline.) We find that naively inputting our attack images to GPT-4V and other VLMs does not work. We thank the reviewers for pointing to this, and we have **added a paragraph addressing this limitation in Section 7**.

We argue, however, that attacks to white-box models are **still of significant importance** for two distinct reasons. Firstly, the existence of such vulnerabilities in open source models suggests similar weaknesses may exist in closed sourced models, even if exposing such vulnerabilities with black-box access requires a different approach. Secondly, a significant number of user-facing applications are and will continue to be built using open-source foundation models, making discoveries such as ours pertinent.

**Have these changes addressed the reviewers’ concerns?** If any concerns remain, please let us know; we will be happy to address them.

---

### Author Response · Authors · 2023-11-21
**Author Response (1/2)**

We thank the reviewers for all of their detailed comments and helpful suggestions. We are glad to hear that the reviewers found our paper “studies a highly pertinent and timely issue” (NiWH) and that our results “demonstrate… the security risks of the VLMs” (DvbG). As we understand it, the reviewers’ main concerns are lack of novelty (C2M9, NiWH, DvbG, dgQe), and application of attacks only to white-box VLMs (NiWH, DvbG, dgQe).

### **Paper Novelty**

To address the concerns about novelty, we have modified our paper accordingly:

1. **We moved a novel method – the prompt matching algorithm – from the appendix into Section 5 of the main paper**, and we applied it to a new disinformation attack. The prompt matching algorithm is a version of our behavior matching algorithm that trains an adversarial image to induce the same model behavior as some target text prompt. To our knowledge, our work is the first to demonstrate this type of attack—[previous work](https://arxiv.org/abs/2307.10490) has failed to get adversarial images to match textual prompts. Using the prompt matching algorithm, we are able to create an image that makes the model believe that the Eiffel Tower is in Rome, an example of model fact editing that could be used to trick VLMs into spreading disinformation to unknowing users. For example, when prompted with an adversarial image, and the text “plan an itinerary for visiting new landmarks in Italy”, the model responds:
    ```
    Day 1: [discussion of the Colosseum, Sistine Chapel, etc.]

    Day 2: … Take a day trip to the Eiffel Tower, now located in Rome, and enjoy the panoramic views of the city from the top. …
    ```
   The prompt matching algorithm is designed to mimic behaviors that are hard to specify with a training dataset. We are able to make the model respond as if the Eiffel Tower is in Rome **without training on any data hand-crafted about the Eiffel Tower or Rome**.

2. In Table 1, we have added GCG text baseline results for the specific string and leak context attacks. We find that this state-of-the-art text-based attack achieves 13.5% and 0% success rate on the specific string and leak context attacks, respectively. Thus, state-of-the-art text-based attacks **dramatically underperform image-based attacks**. In fact, we are completely unable to perform the leak context attack with the GCG text algorithm, while we can achieve up to 96% success with image-based attacks. We note that these text-based attacks are entirely unconstrained, and yet are significantly outperformed by images with 8/255 l-infinity constraint across attack types. **We are the first paper** to show text-based attacks underperform image based attacks across such a wide range of attacks. Our results demonstrate that the addition of a vision input opens up VLMs to **unique and more powerful attacks** such as the leak context attack.

3. In Appendix D we have included the following table that illustrates how our work is novel compared to concurrent papers that also explore adversarial attacks to VLMs:

|                                  | Specific String | Leak Context | Toxic Generation | Jailbreak | $\ell_p$ Constraint | Patch Constraint | Broad Text Baselines | Agnostic to Text Prompts | Text Prompt Matching |
|----------------------------------|-----------------|--------------|------------------|-----------|---------------------|------------------|----------------------|--------------------------|----------------------|
| Carlini et al. (2023)            | ✗               | ✗            | ✓                | ✗         | ✓                   | ✗                | ✗                    | ✓                        | ✗                    |
| Qi et al. (2023)                 | ✗               | ✗            | ✓                | ✓         | ✓                   | ✗                | ✗                    | ✓                        | ✗                    |
| Zhao et al. (2023)               | ✗               | ✗            | ✗                | ✗         | ✓                   | ✗                | ✗                    | ✓                        | ✗                    |
| Shayegani et al. (2023)          | ✗               | ✗            | ✓                | ✓         | ✓                   | ✗                | ✗                    | ✗                        | ✗                    |
| Bagdasaryan et al. (2023)        | ✓               | ✗            | ✗                | ✗         | ✗                   | ✗                | ✗                    | ✓                        | ✗                    |
| Schlarmann & Hein (2023)         | ✓               | ✗            | ✗                | ✗         | ✓                   | ✗                | ✗                    | ✗                        | ✗                    |
| **Ours**                         | ✓               | ✓            | ✓                | ✓         | ✓                   | ✓                | ✓                    | ✓                        | ✓                    |

---

### Author Response · Authors · 2023-11-23
**Author Response (3/2)**

**We have added black-box transferability experiments to Appendix E**. In particular, we test two transferability setups. First, we train an image hijack as before on just one model. Then, we test if directly testing the image on a different model works as an attack. We find that this naive transfer attempt achieves 0% success rate. Next, we try training an image hijack on an ensemble of two models (a technique that [Zou et al](https://arxiv.org/abs/2307.15043). found to be effective for jailbreak transfer of text attacks). Then, we test the image hijack on a third heldout model. We find that **we can train an image hijack on two white-box models at once**, suggesting there exist image hijacks that serve as adversarial inputs to multiple VLMs at once. However, the resulting image has a 0% success rate on the heldout model. Nevertheless, the loss on the held out model decreases, suggesting that further improvements (such as training on a larger ensemble) might make transfer possible in future work.

---

### Meta-Review · Area_Chair_mRiu · 2023-12-08

**Metareview:**

This submission has garnered borderline to negative feedback from reviewers, predominantly due to the perceived lack of transferability of its methods. Although concerns were raised about the novelty of the work in comparison to existing concurrent studies, these were not central to the final decision-making process, in line with ICLR review guidelines.

A notable aspect of this paper is the introduction of a new attack method, 'prompt matching'. However, this concept currently lacks coherent integration into the manuscript, affecting the overall clarity and impact of the paper.

The primary issue influencing the leaning towards rejection is the limited transferability of the adversarial images proposed. This limitation significantly restricts the broader applicability and potential impact of the research.

Considering these factors, it is recommended that the authors focus on revising the paper to enhance its organization, particularly in articulating the 'prompt matching' attack more clearly. Addressing the transferability issues could also enhance the paper's relevance and impact. Such improvements would be beneficial for resubmission in future rounds, potentially elevating the paper's contribution to the field.

**Justification For Why Not Higher Score:**

The recommendation for this paper, leaning towards rejection, is primarily influenced by two key factors: the limited transferability of the proposed adversarial images and the lack of coherent integration of the novel 'prompt matching' attack method into the manuscript.

Limited Transferability: The proposed adversarial images, a central element of this research, demonstrate limited transferability across different settings. This restriction significantly undermines the broader applicability and potential impact of the research, a crucial aspect for high-scoring submissions at ICLR.

Lack of Coherent Integration of Novel Method: While the paper introduces 'prompt matching' as a novel attack mechanism, this concept is not seamlessly woven into the narrative of the manuscript. The lack of clear, organized presentation detracts from the potential significance of this new method, which is essential for a higher scoring paper.

**Justification For Why Not Lower Score:**

N/A

---

### Decision · Program_Chairs · 2024-01-16

Reject